# Natural Product Citronellal can Significantly Disturb Chitin Synthesis and Cell Wall Integrity in *Magnaporthe oryzae*

**DOI:** 10.3390/jof8121310

**Published:** 2022-12-16

**Authors:** Ai-Ai Zhou, Rong-Yu Li, Fei-Xu Mo, Yi Ding, Ruo-Tong Li, Xue Guo, Ke Hu, Ming Li

**Affiliations:** 1Institute of Crop Protection, Guizhou University, Guiyang 550025, China; 2College of Agriculture, Guizhou University, Guiyang 550025, China; 3The Provincial Key Laboratory for Agricultural Pest Management in Mountainous Region, Guiyang 550025, China

**Keywords:** *magnaporthe oryzae*, citronellal, cell wall, physiology, cytotoxicity

## Abstract

Background: Natural products are often favored in the study of crop pests and diseases. Previous studies have shown that citronellal has a strong inhibition effect on *Magnaporthe oryzae*. The objective of this study was to clarify its mechanism of action against *M. oryzae*. Results: Firstly, the biological activity of citronellal against *M. oryzae* was determined by direct and indirect methods, and the results show that citronellal had a strong inhibition effect on *M. oryzae* with EC_50_ values of 134.00 mg/L and 70.48 μL/L air, respectively. Additionally, a preliminary study on its mechanism of action was studied. After citronellal treatment, electron microscopy revealed that the mycelium became thin and broken; scanning electron microscopy revealed that the mycelium was wrinkled and distorted; and transmission electron microscopy revealed that the mycelium cell wall was invaginated, the mass wall of mycelium was separated, and the organelles were blurred. The mycelium was further stained with CFW, and the nodes were blurred, while the mycelium was almost non-fluorescent after PI staining, and there was no significant difference in the relative conductivity of mycelium. In addition, chitinase was significantly enhanced, and the expression of chitin synthesis-related genes was 17.47-fold upregulated. Finally, we found that the efficacy of citronellal against the rice blast was as high as 82.14% according to indoor efficacy tests. Conclusion: These results indicate that citronellal can affect the synthesis of chitin in *M. oryzae* and damage its cell wall, thereby inhibiting the growth of mycelium and effectively protecting rice from rice blasts.

## 1. Introduction

Rice is a famous high-yielding crop providing people with basic daily dietary needs [1]. However, diseases and pests are important factors determining the yield and quality of rice [2,3]. Rice blasts caused by *Magnaporthe oryzae* (*M. oryzae*) can occur in all growth stages of rice and reduce rice yield by up to 50% in severe cases [4]. Currently, chemical control is the most common solution to this problem, and commonly used chemical fungicides include isoprothiolane, tricyclazole, pyraclostrobin, etc. [5,6,7]. Therefore, as an emphasis on biological control, many researchers have turned their attention to natural products and the development of organic agriculture has also increased the demand for biological pesticides [8,9,10,11].

At present, many natural products with antifungal activity have been studied, such as citral, camptothecin, and citronellal, which can effectively inhibit *M. oryzae* [12,13,14,15]. Magnolol and eugenol also have excellent antifungal activity against *Rhizoctonia solani* [16,17]. Citronellal (Figure 1) is a volatile colorless-to-yellowish liquid, with lemon, citronella and rose aroma; soluble in ethanol and the most non-volatile oil; slightly soluble in volatile oil and propylene glycol; and insoluble in glycerin and water. It is naturally present in citronella oil and eucalyptus oil, which is widely used in medicine, edible spices, and agriculture [18]. For example, citronellal can inhibit the expression of leukotrienes, reduce inflammation, reduce edema, anti-inflammatory and analgesic [19,20], and it is an effective mosquito repellent [21]. It can also be used to add flavor to food and beverages and enrich the aroma of products, such as synthetic menthone, menthol, citronellol, hydroxycitronellal, and other edible incense materials [22]. Due to its antifungal properties, citronellal can improve the hypersensitivity of fungi to membrane interferents, reduce ergosterol levels, and damage cell membrane stability, especially against *Candida albicans* [23]. Citronellal has a dual antifungal mechanism, including cell membrane destruction, in addition to mitochondrial and DNA oxidative damage [24]. Our previous studies found that citronellal has a good inhibitory effect on *M. oryzae*. In order to make its use in agricultural production more widespread, our main objective is to clarify the action mechanism of citronellal on *M. oryzae*. Thus far, the mechanism of citronellal against *M. oryzae* has not been researched. 

The structure of this study is as follows. We observed the effects of citronellal on mycelial morphology, cell microstructure, and cell wall membrane in *M. oryzae*. The changes in membrane permeability and various biochemical substrate levels were detected from a physiological point of view, and expression levels of related genes regulating chitin synthesis were determined on a molecular level. The control effect of citronellal on rice blast was determined through laboratory experiments. The ultimate goal of understanding these mechanisms is not only to clarify the action mechanism of citronellal on *M. oryzae* and its application prospects, but also lay a theoretical foundation for studying the application of citronellal in production and better designing and synthesizing new compounds against *M. oryzae*.

## 2. Materials and Methods

### 2.1. Magnaporthe oryzae, Medium, Rice Leaves and Citronellal

*M. oryzae* was isolated from rice blast disease, provided by Institute of Crop Protection, Guizhou University. Preparation of agarized potato medium: wash and peel the potato, weigh 200 g and cut into pieces, add 1000 mL of water and boil for 20 min, filter, fix the volume to 1000 mL, add 20 g of glucose and 20 g of agar to dissolve and divide, autoclave at 121 °C for 20 min, and use. The non-agarized potato medium was based on the agarose potato medium without agar. Leaves of 5-instar rice collected from field susceptible varieties. The temperature range for rice growth was from 22 to 32 °C and the humidity range was from 50 to 90%. Citronellal with a purity of 96% was purchased from Shanghai Aladdin Biochemical Technology Co., Ltd. (Shanghai, China) and stored at 4 °C.

### 2.2. Antifungal Activity of Citronellal on Mycelial Growth

The mycelial growth rate method suggested by Mo et al. [16] was followed. Citronellal was dissolved with N, N-dimethylformamide, diluted with water to 5 concentrations, and evenly mixed with PDA. *M. oryzae* colonies with a diameter of 5 mm were placed in the medium and cultured at 28 °C for 9 d under dark conditions. The EC_50_ (concentration for 50% of maximal effect) of citronellal was calculated by SPSS 25.0 [25].

The indirect activity method was slightly modified with reference to Soylu et al. [26]: Melted PDA medium was poured into the plates: 10 mL per plate. *M. oryzae* colonies with diameters of 5 mm were placed in the center of the plate, and the plate was turned upside down. Aspirated 0.64, 1.92, 3.20, 4.48, 5.76, 7.04, 8.32, 73.6 μL of citronellal essence were dropped in the center of each plate lid with no drops of fungicide, and then used as a control group (CK). Plates were immediately sealed with parafilm, incubated at 28 °C under dark conditions, and colony diameter was measured after 9 d.

### 2.3. Determination of the Mycelial Quantity of M. oryzae

The method suggested by Mo et al. [16] was followed. *M. oryzae* colonies with a diameter of 5 mm were placed into the non-agarized potato media, and shaken at 28 °C and 150 r/min for 72 h, and 5 g of mycelium was added to 100 mL of citronellal-containing the non-agarized potato media. Culturing was continued under the same conditions: mycelia were filtered out, washed with distilled water 3 times, and vacuum filtered for 10 min, and the remaining weights of mycelia and loss rates were calculated according to the following Equation (1).
(1)$$\mathrm{Loss}\;\mathrm{rate}\;\left(\%\right)=\frac{\mathrm{Initial}\;\mathrm{weight}\;\mathrm{of}\;\mathrm{mycelia}-\mathrm{Final}\;\mathrm{weight}\;\mathrm{of}\;\mathrm{mycelia}}{\mathrm{Initial}\;\mathrm{weight}\;\mathrm{of}\;\mathrm{mycelia}}$$

### 2.4. Observation of Mycelial Morphology

The method was slightly modified with reference to He et al. [27] Citronellal was dissolved with N, N-dimethylformamide, then diluted with water to three concentrations of EC_25_, EC_50_, and EC_75_, and evenly mixed with PDA. *M. oryzae* colonies with diameter of 5 mm were placed in the medium. Aseptic cover slides were placed around the colony and cultured at 28 °C under dark conditions. The morphologies of the mycelia were observed under a light microscope after 36, 48, 72, 96, and 120 h.

### 2.5. Mycelia Morphology and Ultrastructure of Mycelial Cell

The method of SEM and TEM suggested by Mo et al. [16] and Li et al. [12,13] was followed. The mycelia were cultured according to the method described in Section 2.4. A small amount of mycelium was taken and placed in 2.5% glutaraldehyde–0.2 M phosphate-buffered solution, and stored at 4 °C for 12 h. The fixative solution was poured out, and mycelia were rinsed 3 times with 0.1 M, pH 7.0 phosphate-buffered solution for 15 min each time, and then fixed with 1% osmic acid solution for 2 h. The mycelia were removed and rinsed 3 times with 0.1 M, pH 7.0 phosphate buffer for 15 min each time; dehydrated with 30%, 50%, 70%, 80%, 90%, and 95% ethanol solution for 15 min at each concentration; and then treated with anhydrous ethanol for 2 times for 20 min each time. The mycelia were removed for further treatment under scanning electron microscopy (SEM) and transmission electron microscopy (TEM).

SEM: The mycelia were treated with a mixture of ethanol and isoamyl acetate (*v/v* = 1/1) for 30 min, followed by 1 h treatment with pure isoamyl acetate. The treated samples were subjected to critical point drying and coating before observations.

TEM: The mycelia were placed in acetone for 20 min, and then treated with a mixture of embedding agent and acetone (*v/v* = 1/1) for 1 h, followed by a mixture of embedding agent and acetone (*v/v* = 3/1) for 3 h, and then pure embedded agent for 12 h. The permeabilized mycelia were embedded at 70 °C for 12 h and sectioned at 70~90 nm using a Leica EMUC7 ultrathin microtome. The slices were stained with lead citrate solution and uranium oxyacetate 50% ethanol saturated solution for 5~10 min, respectively. The slices were observed under TEM after drying.

### 2.6. Scanning Cell Membrane and Cell Wall Integrity

Propidium iodide (PI) staining and Calcofluor white (CFW) staining followed the methods of Li et al. [12,13] and Mo et al. [16] The fluorescent dye PI (propidium iodide) is an analogue of ethidium bromide that releases a red fluorescence when embedded in double-stranded DNA. It can reflect the damage of the cell membrane [28]. CFW is a non-specific brightening dye that binds to β-linked glycosides to produce fluorescence. Staining can reflect damage to the cell wall [29]. The mycelia were cultured according to the method described in Section 2.4.

Propidium iodide (PI) staining: Mycelia were removed and washed 3 times with distilled water. A small number of mycelia were taken, and 500 μL of PBS buffer solution (50 mmol/L pH 7.0) was added. Then, 5 μL of PI (produced by Beijing Solaibao Technology Co., Ltd., Beijing, China) was added and shaken well; the specimen was placed in a water bath at a constant temperature of 30 °C for 15 min, washed 3 times with a 50 mmol/L pH 7.0 PBS buffer solution, placed on a glass slide, and observed under a fluorescence microscope with an excitation wavelength of 535 nm.

Calcofluor white (CFW) staining: The mycelia were washed 3 times with distilled water. A small number of mycelia were collected, and 5 μL of HEPENGBIO staining solution (Hepeng Biological Technology Co., Ltd., Shanghai, China) was added, followed by full shaking and incubation in the dark for 5 min at room temperature. Next, 500 μL of HEPENGBIO cleaning solution was added, and this step was repeated 5 to 7 times. Then, 100 μL of HEPENGBIO cleaning solution was added, mixed gently and 10 μL of mycelium suspension was removed. The solution was placed on a glass slide, covered with a coverslip, and observed under a fluorescence microscope with an excitation wavelength of 535 nm and emission wavelength of 440 nm. The cell wall had a strong blue fluorescence.

### 2.7. Determination of Mycelial Cell Membrane Permeability

The relative conductivity is an indicator of cell membrane permeability [30]. *M. oryzae* colonies with a diameter of 5 mm were placed in the PDB medium and incubated in a shaker at a temperature of 28 °C and a speed of 150 r/min for 72 h. Citronellal was added to mycelia suspensions at a ratio of 9:1 (concentration of citronellal in the medium reached EC_25_, EC_50_, and EC_75_), and sterile water was used as a control, then mixed thoroughly. The electrical conductivity of the mycelia was measured at 30, 60, 90, 120, 240, 360, 520, and 680 min with a DDS-307 conductivity meter. After 680 min, the samples were placed in a boiling water bath for 10 min, cooled, and the measured conductivity was the same a the absolute conductivity. Each treatment was repeated three times, and the average value was taken to calculate the relative permeability for each period time, according to the following Equation (2):(2)$$\mathrm{Relative}\;\mathrm{conductivity}\;\mathrm{at}\;\mathrm{each}\;\mathrm{time}\;\left(\%\right)=\frac{\mathrm{Electrical}\;\mathrm{conductivity}\;\mathrm{at}\;\mathrm{each}\;\mathrm{time}}{\mathrm{Absoluteel}\;\mathrm{ectrical}\;\mathrm{conductivity}}\times 100$$

### 2.8. Determination of Mycelial Chitinase Activity

Chitinase is an enzyme that catalyzes the hydrolysis of chitin [31]. The hydrolysis of titin by chitinase produced N-acetylglucosamine, which further produced a red compound with p-dimethylaminobenzaldehyde, with a characteristic absorption peak at 585 nm, and the rate of increase in the absorption value reflected the activity of chitinase. The mycelia were cultured according to the method described in Section 2.7. After continuous cultivation for 9, 24, 48, 72, 96, 120, and 144 h, mycelia were washed with distilled water 3 times, filtered for 10 min. Then, 0.15 g was added to 1.5 mL chitinase extract solution (produced by Beijing Solaibao Technology Co., Ltd., Beijing, China) and homogenized in an ice bath with centrifuged at 10,000 *g* and 4 °C for 20 min. Then, the supernatant was put on ice for measurement. Chitinase standard curve was drawn, and chitinase activity was measured (chitinase was obtained from Beijing Solaibao Technology Co., Ltd., Beijing, China).

### 2.9. Determination of Mycelial Β-1,3-Glucanase Activity

β-glucanase can effectively break down β-glucan in the cell wall [32]. Kombucha polysaccharide was hydrolyzed by β-1,3-GA with endo-cutting of β-1,3-glucosidic bonds to produce reducing ends, and its enzymatic activity was calculated by measuring the rate of reducing sugar production. Mycelia were cultured according to the method described in Section 2.7. After continuous cultivation for 9, 24, 48, 72, 96, 120, and 144 h, the mycelia were washed with distilled water 3 times, filtered for 10 min. Next, 0.10 g was added to 1.0 mL β-1,3-glucanase extract solution (produced by Beijing Solaibao Technology Co., Ltd., Beijing, China) and homogenized in an ice bath with centrifuge at 12,000 *g* and 4 °C for 10 min. Then, the supernatant was put on ice for measurement. The β-1,3-glucanase standard curve was drawn, and β-1,3-glucanase activity was measured (produced by Beijing Solaibao Technology Co., Ltd., Beijing, China).

### 2.10. Detection of Expression of Mycelial Chitinase Gene

The method suggested by Song et al. [33] was followed. The mycelia were cultured according to the method of step 2.7. After 72 h continuous cultivation, mycelia were washed with distilled water 3 times, filtered for 10 min, and 0.1 g mycelia was weighed for subsequent experiments.

Total RNA extraction: The total RNA for RT-qPCR analysis was extracted using RNAprep Pure Plant Kit. (Beijing Tiangen Biochemical Technology Co., Ltd., Beijing, China). The yield and purity of RNA samples were quantified using NanoDrop ND-1000 (NanoDrop, Wilmington, DE, USA). Additionally, a Gel Electrophoresis Imaging System was used to detect the quality of RNA.

cDNA synthesis: 2.0 µg of RNA was taken for reverse transcription with a FastKing gDNA Dispelling RT SuperMix (Beijing Tiangen Biochemical Technology Co., Ltd., Beijing, China) in a 20 µL reaction volume according to the manufacturer’s instructions.

Quantitative real-time PCR analysis: The chitinase-related genes of the M. oryzae were queried, and 15 gene names were clarified: *MGG*_05533, *MGG*_04732, *MGG*_01247, *MGG*_01336, *MGG*_01876, *MGG*_08054, *MGG*_03599, *MGG*_00086, *MGG*_07927, *MGG*_04073, *MGG*_10333, *MGG*_04534, *MGG*_06594, *MGG*_11231, and *MGG*_08458. Through the Rice Blast Genome Library (http://fungi.ensembl.org/Magnaporthe_oryzae/Info/Index, accessed on 16 December 2022), the sequences of these 15 genes were found. The actin gene was used as an internal control. Gene-specific primer pairs were designed using the Sangon primer design and synthesis online platform (https://www.sangon.com/newPrimerDesign, accessed on 16 December 2022) on the basis of the transcript sequence and synthesized by Sangon Biotech Co., Ltd., Shanghai, China) (Table 1). The PCR was performed using SYBR Green’s method and a Bio-Rad CFX96 real-time PCR system (Bio-Rad, Berkeley, CA, USA) with a total of 96-well plates; each sample was set up with three biological replicates. The final volume for each reaction was 20 µL with the following components: 0.8 µL of cDNA template (1 ng/µL), 10 µL of 2× SG Fast qPCR Master Mix (Sangon, Shanghai, China), 0.6 µL of forward primer (200 nM), 0.6 µL of reverse primer (200 nM), and 8 µL of PCR-grade water. The reaction was conducted under the following conditions: 95 °C for 1 min, followed by 40 cycles of denaturation at 95 °C for 5 s, annealing at 56 °C for 10 s, and extension at 72 °C for 15 s. The melting curve was obtained by heating the amplicon from 65 °C to 95 °C at increments of 0.5 °C per 5 s. The relative quantification of gene expression was computed using the 2^−∆∆Ct^ method.

### 2.11. Detection of Expression of Mycelial Β-1,3-Glucanase Gene

The mycelia were cultured according to the method of step 2.7. Total RNA extraction, cDNA synthesis, quantitative real-time PCR analysis, Gene sequence search, primer design (Table 2), and synthesis were carried out according to the method described in Section 2.10. The genes related to β-1,3-glucanase of *M. oryzae* were queried, and 17 gene names were identified: *MGG*_00659, *MGG*_06722, *MGG*_06512, *MGG*_07846, *MGG*_01001, *MGG*_07331, *MGG*_03208, *MGG*_08370, *MGG*_10400, *MGG*_04689, *MGG*_14087, *MGG*_09995, *MGG*_17486, *MGG*_11861, *MGG*_09619, *MGG*_10591, and *MGG*_00865.

### 2.12. Evaluation of Indoor Effectiveness

To test whether citronellal protects rice plants against rice blast disease, we used the laboratory leaf test method [34]. Healthy leaves of the same leaf age were selected. The middle part of the leaf, about 10 cm, was cut and soaked in 75% alcohol for 10 s, and then washed with sterile water 3 times. Citronellal was prepared with concentrations of EC_25_, EC_50_ and EC_75_, and rice leaves were soaked in citronellal for 10 s. Then, three small wounds were gently cut at equal intervals on the leaf surface with a sterilized needle and placed in an inoculation box covered with moisturizing cotton. Mycelium blocks of 5 mm diameter were inserted into each small wound. Inoculated plants were kept in a growth chamber at 28 °C with 80% humidity and darkness for the first 24 h, followed by a 12/12 h light (simulating natural sunlight in 5000 LUX)/dark cycle. The incidence of rice leaves was counted after 14 d. A rating scale was used to measure the disease incidence and severity [35] 0, no lesions; 1, small brown spots the size of a pinhead; 2, large brown spots; 3, oval necrotic gray spots with brown margins about 1–2 mm in diameter; 4, typical spots, oval, 1–2 cm long, usually confined to the main veins of the leaf, covering less than 2% of the entire leaf area; 5, typical spots, with spots covering 10% of the whole leaf area; 6, typical spot, spot area of 10–25% of the whole leaf area; 7, typical spot, spot area of 26–50% of the whole leaf area; 8, typical spot, spot area of 51–75% of the whole leaf area, many leaves die; 9, all leaves die. The disease index was calculated according to the following Equations (3) and (4). These and other data were subjected to analysis of variance (ANOVA) using SPSS 25.0 software (SPSS Inc., Chicago, IL, USA).
(3)$$\mathrm{Disease}\;\mathrm{index}\;\left(\%\right)=\frac{\sum \left(\mathrm{Rice}\;\mathrm{blast}\;\mathrm{rating}\;\times \mathrm{Number}\;\mathrm{of}\;\mathrm{leavesat}\;\mathrm{that}\;\mathrm{rating}\right)}{\left(\mathrm{Total}\;\mathrm{rice}\;\mathrm{leaves}\;\times 9\right)}\times 100$$
(4)$$\mathrm{Control}\;\mathrm{effect}\;\left(\%\right)=\frac{\;(\mathrm{control}\;-\mathrm{treatment})}{\mathrm{control}}\times 100$$

## 3. Results

### 3.1. The Inhibition of Citronellal on Mycelial Growth of M. oryzae

The inhibitory effect of citronellal against *M. oryzae* was determined by contact and indirect activity methods (Figure 2). The results show that citronellal has a good inhibition effect on *M. oryzae* with EC_25_, EC_50_, and EC_75_ values of 85 mg/L, 134 mg/L, and 218 mg/L, respectively. The experimental results of the indirect activity method (Figure 2B) show that the colony diameter of *M. oryzae* became smaller and smaller with increasing concentrations of citronellal. When the concentration of citronellal reached 150 μL/L, there was almost no trend of mycelial growth, with an EC_50_ value of 70.48 μL/L. It was shown that citronellal also had a better indirect activity inhibition effect on *M. oryzae*.

### 3.2. Effect of Citronellal on Mycelial Quantity of M. oryzae

As shown in Figure 3, after 72 h, the mycelium weight of the control groups was increased by 35.83%, while the mycelium weights of citronellal groups (85 and 134 mg/L) were significantly different from the control group with an increase of 1.15% and a decrease of 5.89%, respectively. However, the weight loss rate of mycelia significantly increased after treatment with a high content of citronellal with a loss rate of 88.59%. The results show that citronellal could inhibit the growth of mycelia in *M. oryzae* and reduce the weight of *M. oryzae*.

### 3.3. Effect of Citronellal on Mycelial Morphology of M. oryzae

As shown in Figure 4, the mycelia of *M. oryzae* were damaged to varying degrees after treatment with different concentrations of citronellal at different times. The mycelium morphology of the control groups did not significantly change, with longer mycelia and a clear separation at different times. The mycelium morphology showed almost no difference between treatment groups before 36 h, and then, mycelium began to thin and deform over time. In particular, the mycelium of *M. oryzae* appeared vacuolated, broken, and shattered into pieces, and twisted after 72 h of treatment with citronellal. This phenomenon became increasingly evident as the citronellal increased. It suggested that citronellal may affect the growth of the pathogen.

### 3.4. Effect of Citronellal on Mycelial Morphology and Ultrastructure of M. oryzae

Under a scanning electron microscope, in the control groups (Figure 5A,B), the mycelia of *M. oryzae* were uniform in thickness. The mycelial surfaces of *M. oryzae* were smooth, plump, and clearly separated. In the citronellal groups (85 mg/L, Figure 5C,D), the mycelial surfaces of *M. oryzae* were slightly shrunken, swelled, and uneven in thickness. In Figure 5E,F, the mycelium surface of *M. oryzae* was partially deformed, partially ruptured, and wrinkled. As shown in Figure 5G,H, the mycelial surfaces of *M. oryzae* were abnormally enlarged, distorted, and even broken. It can be seen that different concentrations of citronellal damaged mycelial shape and structural integrity.

The ultrastructures of *M. oryzae* under different treatments, as observed under a transmission electron microscope, are shown in Figure 6. The mycelium ultrastructures of *M. oryzae* in the control group had regular cell morphologies, uniform cell wall textures, uniform thicknesses closely linked with the cell membrane, organelles evenly distributed in the cytoplasm, and clear and complete structures (Figure 6A,B). In the citronellal groups (Figure 6C,D), cells were slightly deformed and slightly vacuolated, and the distribution of intracellular substances was uneven. Following the exposure to 134 mg/L citronellal (Figure 6E,F), cells were invaginated and deformed. Obvious swelling, cytoplasmic wall separation extravasation, and many vesicular structures were observed in the cells, and there was even no complete organelle structure. When hyphae were exposed to 218 mg/L citronellal (Figure 6G,H), the cell wall of *M. oryzae* was severely lysed, while the cell membrane was invaginated. The plasmon of *M. oryzae* was clearly separated, and the intracellular structure was blurred with content extravasation. This indicated that the action site of citronellal against *M. oryzae* might be the cell wall of mycelium.

### 3.5. Mycelial PI and CFW Fluorescence Reaction of M. oryzae

PI is a nucleic acid dye that cannot penetrate intact cell membranes. Necrotic cells have a strong fluorescence due to loss of membrane integrity, where the dye enters the cell and binds to DNA, while intact cell membranes are not damaged since the dye cannot penetrate into the cell and bind to DNA, and thus, it does not show fluorescence. Based on this feature, dead cells can be identified using PI staining, and the more significant the fluorescence, the more serious the cell membrane damage. As shown in Figure 7A,B, fluorescence could be barely observed in all the treatments, and there was no significant difference in the percentage of dead cells after fluorescence quantification. The results show that the fluorescence of mycelia hardly changes with the increase in citronellal concentration, indicating that citronellal did not damage the membrane integrity of *M. oryzae*.

CFW is a non-specific fluorescent whitening dye that binds to cellulose and chitin at the cell wall to generate fluorescence, detecting changes in the formation of the cell wall, which has a strong blue fluorescence. As shown in Figure 8, all treated mycelia of *M. oryzae* had a blue fluorescence after 72 h. The fluorescence intensity of treated mycelium in nodes separation became increasingly blurred with the increase in citronellal content, while control mycelial nodes emitted a strong, bright and clear fluorescence, indicating that citronellol could destroy the integrity of the cell wall in *M. oryzae*.

### 3.6. Effect of Citronellal on Mycelial Cell Membrane Permeability of M. oryzae

Relative conductivity is an important index for measuring the permeability of the cell membrane. The relative conductivity of the mycelia in all the treatments steadily increased to 30–120 min (Figure 7C). After 120 min, the relative conductivities of the mycelia in low, medium, and high concentrations of citronellal treatments and control groups all increased from about 20%. The relative conductivity of the mycelia in all treatments significantly increased until reaching 360 min. There was a significant difference between the citronellal treatment group (218 mg/L) and other treatment groups, the relative conductivity of the treatment group (218 mg/L) at this time was 49.93%, and there were no significant differences between the other three groups (the relative conductivities are 34.78%, 37.65%, and 41.61%, respectively). These results further confirm that citronellal did not directly damage the cell membrane of the mycelium.

### 3.7. Effect of Citronellal on Chitinase Activity in M. oryzae

Chitin is an important component of the cell wall, and chitinase is an important indicator to evaluate its integrity. Absorbance was measured at 540 nm using a visible spectrophotometer, and there was a good linear relationship between the chitinase concentration of standard solvent and the area of the absorbed peak. The regression equation was Y = 0.33X − 0.2074, and the correlation coefficient of the standard curve was 0.9994 (Figure 9A). As shown in Figure 9C, the chitinase activity of mycelium in treatments with citronellal began to gradually increase after 24 h, reached a peak at 72 h, and then began to decline. While the chitinase activity of the mycelium in the control group did not significantly change with time, it fluctuated below 5 U/g, and remained lower than the overall level of treatment groups. Especially, the chitinase activities of the mycelia treated with 85 mg/L, 134 mg/L, and 218 mg/L citronellal were 42.43, 54.57, and 55.28 times higher than those of the control group after 72 h, respectively. This enhanced chitinase activity resulted in accelerated hydrolysis of chitin in the cell wall of *M. oryzae*, indicating that citronellal indirectly acted at the cell wall of *M. oryzae* and accelerated the death of *M. oryzae* cells.

### 3.8. Effect of Citronellal on Β-1,3-Glucanase Activity of M. oryzae

Glucan is an important component of the cell wall. β-1,3-glucanase can catalyze its hydrolysis. The absorbance was measured at 540 nm using a visible spectrophotometer, and standard curve R^2^ = 0.9991, Y = 0.5322X − 0.0635 was obtained by using β-1,3-glucanase concentration as a horizontal coordinate and the absorbance value as the vertical coordinate (Figure 9B). As shown in Figure 9D, for the control group mycelia and those treated with a medium concentration (134 mg/L) for 9 h, enzyme activity was higher than at low-concentration (85 mg/L) and high-concentration (218 mg/L) citronellal treatments. It first showed a decreasing trend, and then started to increase after 48 h, whereas the enzyme activity of mycelium had an increasing trend from a lower value at 9 h treatment with a low and high concentration of citronellal. The enzyme activity of mycelium in all the treatments reached a peak at 72 h, decreasing and stabilizing after 96 h. The difference in enzyme activity of mycelium in all the treatments was greatest after 48 h. The β-1,3-glucanase activity of the mycelium treated with citronellal (218 mg/L) was only 1.96 times higher than that in the control group. The overall trend of the treated groups was higher than the control group, but the difference was not significant, indicating that citronellal had no significant effect on β-1,3-glucan of *M. oryzae*.

### 3.9. Effect of Citronellal on Chitinase Gene Expression in M. oryzae

To further verify the regulatory effect of citronellal on the chitin synthesis pathway of *M. oryzae*, we selected 15 chitinase genes of *M. oryzae* for quantitative real-time PCR analysis. As shown in Figure 10A, the results show that expression levels of *MGG*_08054 and *MGG*_04073 were lower than those of the control group after treatment with citronellal for 72 h, and gene expressions were downregulated by 1.41 and 0.09 times, respectively. Meanwhile, the expression levels of the remaining 13 genes were higher than those of the control group and had an upregulation from 1.1790- to 4.1709-fold. Especially, expression levels of *MGG*_04732, *MGG*_01876, and *MGG*_03599 were upregulated by more than three-fold.

### 3.10. Effect of Citronellal on Β-1,3-Glucanase Gene Expression of M. oryzae

In order to further verify the effect of citronellal on the β-1,3-glucanase gene expression of *M. oryzae*, 17 related genes were screened for Quantitative Real-Time PCR analysis. The results show that the expression levels of five genes (*MGG*_01001, *MGG*_10400, *MGG*_04689, *MGG*_09619, and *MGG*_00865) were lower than those of the control treated with citronellal for 72 h and were downregulated by 0.61–1.44-fold (Figure 10B), while expression levels of the other 12 genes were upregulated by 4.09–17.47-fold, which were significantly higher than that of control. Therefore, most β-1,3-glucanase genes are positively regulated when they are transcribed into mRNA, which might affect the subsequent synthesis of β-1,3-glucan.

### 3.11. Indoor Control Effect of Citronellal on Rice Blast

In order to clarify the control effect of citronellal on a rice blast, a laboratory leaf test method was used for determination. The lesions on inoculated leaves gradually decreased, and the symptoms of the yellowing of leaves were alleviated with the increase in citronellal content after 14 d (Figure 11A). After quantifying symptoms, statistical results show that the control effects of treatment groups were significantly different: 14.29%, 25.00% and 82.14%, respectively (Figure 11B), indicating that citronellal had a good control effect on rice blast.

## 4. Discussion

*M. oryzae* can cause rice blast, which poses a serious threat to rice yields worldwide [36,37]. As a result, natural products with a low toxicity to non-target organisms have attracted a widespread attention of researchers [38]. Citronellal, an aromatic component present in essential oils of various plants, such as citronella oil and eucalyptus oil, can be extracted by various routes [39]. It is not only highly effective in mosquitoes, but also has strong antifungal properties against a variety of pathogenic fungi [40,41,42]. Studying the antifungal mechanism of citronellal can provide an important reference for subsequent research and applications and play an important role in agricultural applications.

In this study, the inhibition effect of citronellal was measured from the perspective of contact and indirect activity, and the inhibition mechanism of citronellal against *M. oryzae* was systematically studied from the perspective of morphological structures, physiology and molecular levels. The results show that citronellal had a strong inhibition effect on *M. oryzae*, increased the activity of chitinase, significantly up-regulated the expression level of related genes, and thus accelerated the hydrolysis of chitin. β-1,3-glucanase-related gene expression was significantly upregulated, while glucanase activity was not significantly different from the control group. This may be due to the presence of other related genes regulating the expression of glucanase. This expression inhibited the growth of mycelia and led to the destruction of the cell wall, which was consistent with the distortion and rupture of mycelia observed by SEM and the serious dissolution of cell walls observed by TEM. It was also found that citronellal had no significant effect on the cell membrane permeability of mycelium in *M. oryzae* by relative conductivity measurements, which was consistent with a lack of clear fluorescence in PI staining, and might be related to the fact that citronellal did not directly affect the cell membrane. Therefore, the main pathway of citronellal against *M. oryzae* might be related to genes regulating chitin synthesis, destroying cell walls, damaging cell function, and inhibiting mycelium growth. Thus, citronellal treatment has a significant protective effect on rice leaves. At present, the pharmacodynamic mechanism of citronellal had been widely studied in *Penicillium* and *Candida albicans*. For example, citronellal could reduce ergosterol synthesis in *Penicillium digitorum* spores by inhibiting the expression of *ERG*3, thus affecting spore germination. This impaired the cell membrane integrity of *P. fingertip* by downregulating the *ERG* gene responsible for converting lanolin to ergosterol [43,44], and exerted antifungal properties by inhibiting the biofilm formation of *C. albicans* [23].

The reported studies suggest that citronellal might exert its inhibition effect by affecting the synthesis of ergosterol and biofilm [45,46]. These results are inconsistent with the findings of this study. The possible factors for discrepancy included different pathogenic species or citronellal having multiple sites of action on the fungal cell wall and cell membrane. However, citral, which is also a natural product of aldehydes, was shown to inhibit *Trichoderma viride* [47] and *M. oryzae* [14] by disrupting the cell wall in studies of antifungal mechanisms. Cinnamaldehyde could also exhibit antifungal activity against *Citrus sour rot* and *Fusarium* by interfering with the construction of cell walls [48,49]. When used alone or in combination with cinnamaldehyde, citronellal could inhibit mycelial growth by accelerating the destruction of cell membranes and cell walls. It could also be used to prevent mold and prolong the shelf life of citrus, strawberry and other fruits [50,51,52]. These results support the conclusion of this study: citronellal has the potential for effective applications in the prevention and control of rice blasts.

This study screened genes that regulate chitin synthesis and confirmed the strong inhibitory effect of citronellal on *M. oryzae* and a better control effect on rice blast, exerting an antifungal effect via the synthesis of chitin and disruption of the cell wall. These conclusions could provide a reference for the selection of drugs for the prevention and control of rice blasts and lay the foundation for further study of molecular mechanisms, field applications, and the design and synthesis of target drugs. In particular, the indirect activity of citronellal brings more possibilities to its application. It could also be combined with related technologies to improve the biological activity of plant extracts [53,54,55] to improve its stability and achieve a slow release of the drug. Although this study reveals the antifungal mechanism of citronellal against *M. oryzae* and preliminarily elucidated its control effect on rice blast, its molecular regulation mechanism and field application technology still require further study.

## 5. Conclusions

Currently, rice blast is dependent on several types of chemicals, and thus, the development and application of natural products are extremely urgent. Citronellal has good antifungal activity against *M. oryzae*, and reveals its potential in the prevention and control of rice blasts. In this study, it was found that citronellal could affect the mycelia morphology of *M. oryzae*, destroy the cell wall, and inhibit mycelia growth. To further clarify the action mechanism of citronellal, the relevant indexes were determined to verify. The results show that citronellal up-regulated the expression of chitinase- and β-1,3-glucanase-related genes, and significantly increased chitinase activity. These results suggest that citronellal can upregulate the expression of genes related to chitin synthesis, and interfere with this synthesis, destroying the cell wall, inhibiting mycelia growth, and providing effective protection against rice blast. This study preliminarily clarifies the action mechanism of citronellal on *M. oryzae* and provides an effective method for the green prevention and control of rice blasts and the safe production of rice. However, the key genes in regulatory pathways and their specific regulatory relationships are not yet clear. The instability of citronellal limits its application in this field. Therefore, in the next stage, we aim to pay attention to key genes in the regulatory pathway and clarify their regulatory mechanism. Additionally, thorough research should be combined with related technologies to improve the biological activity of plant extracts and their potential for applications in the field of agriculture.

## Figures and Tables

**Figure 1 jof-08-01310-f001:**
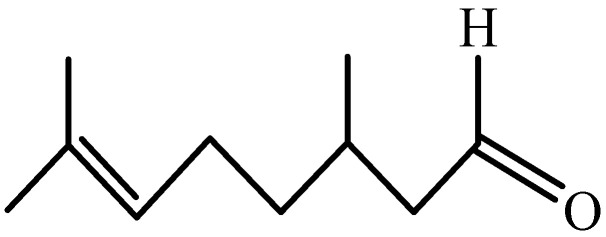
Chemical structure of citronellal.

**Figure 2 jof-08-01310-f002:**
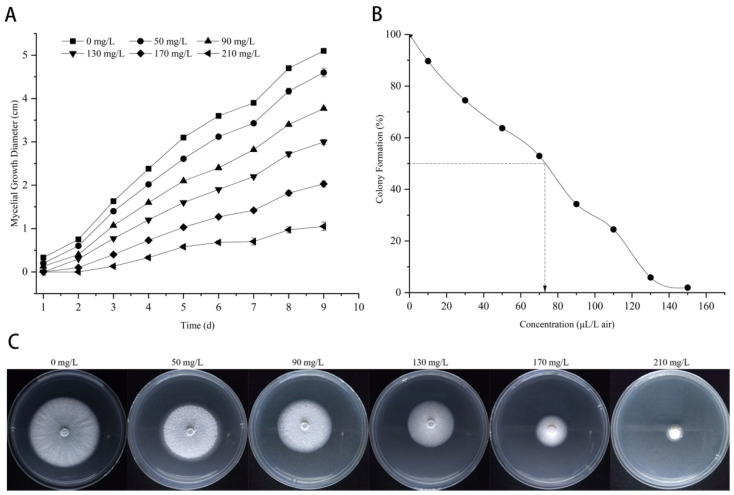
Inhibitory effect of citronellal on *M. oryzae.* Error bars denote standard error of mean for three independent experiments, and below is the same. (**A**) Mycelial growth rate method, (**B**) Indirect activity method, (**C**) The colony formation of *M. oryzae* by mycelial growth velocity method.

**Figure 3 jof-08-01310-f003:**
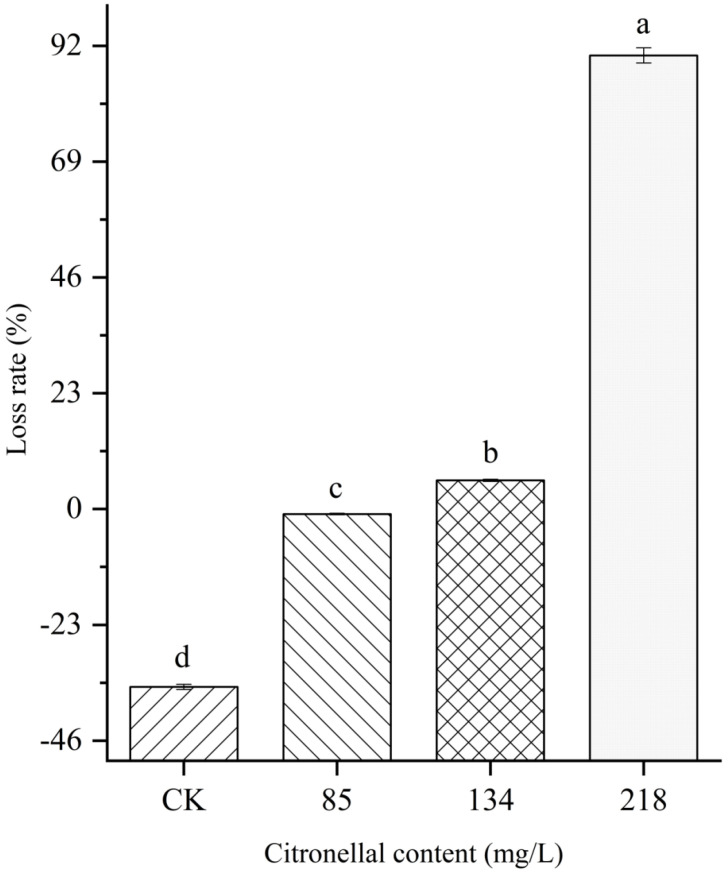
The loss rate of mycelium. Different lowercase letters denote statistically significant differences at α = 0.05; below is the same.

**Figure 4 jof-08-01310-f004:**
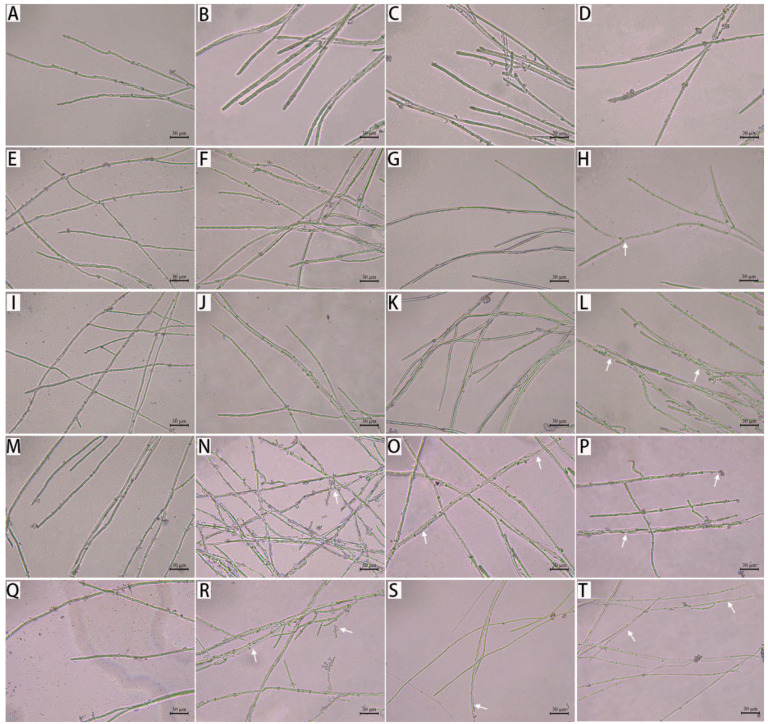
Effects of citronellal on morphology of *M. oryzae:* 36 h: (**A**) (CK), (**B**) (85 mg/L), (**C**) (134 mg/L), (**D**) (218 mg/L); 48 h: (**E**) (CK), (**F**) (85 mg/L), (**G**) (134 mg/L), (**H**) (218 mg/L); 72 h: (**I**) (CK), (**J**) (85 mg/L), (**K**) (134 mg/L), (**L**) (218 mg/L); 96 h: (**M**) (CK), (**N**) (85 mg/L), (**O**) (134 mg/L), (**P**) (218 mg/L); 120 h: (**Q**) (CK), (**R**) (85 mg/L), (**S**) (134 mg/L), (**T**) (218 mg/L). The scale bar is 30 μm.

**Figure 5 jof-08-01310-f005:**
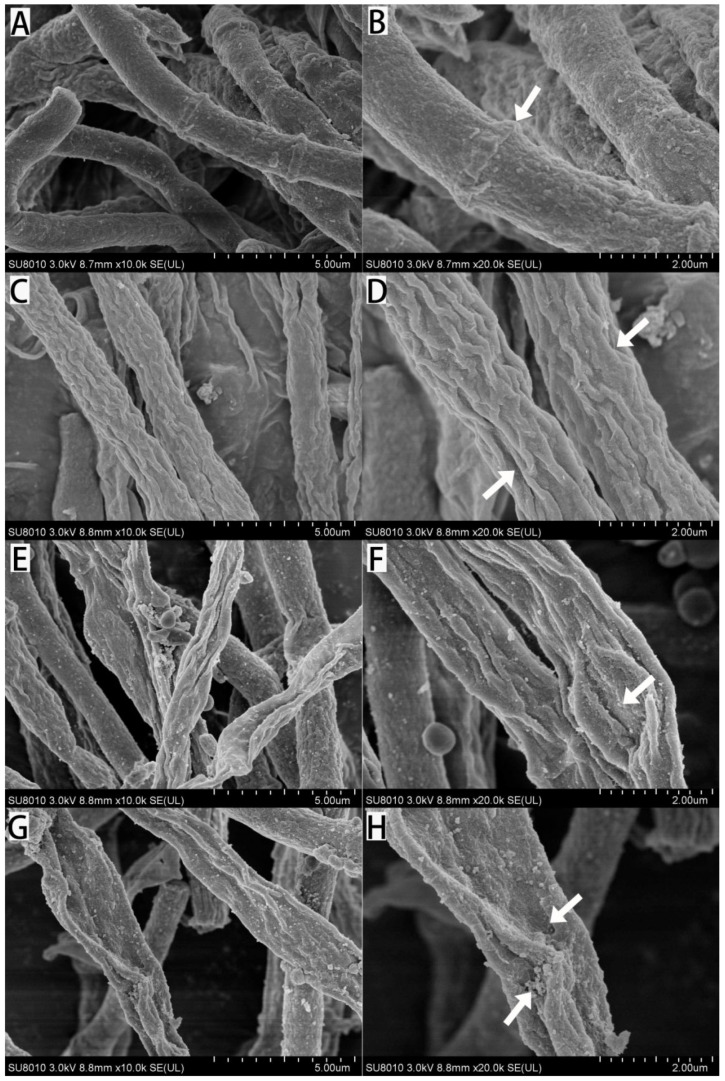
Morphology of mycelium after 72 h treatment with different concentrations of citronellal. (**A**,**B**): CK; (**C**,**D**): 85 mg/L; (**E**,**F**): 134 mg/L; (**G**,**H**): 218 mg/L.

**Figure 6 jof-08-01310-f006:**
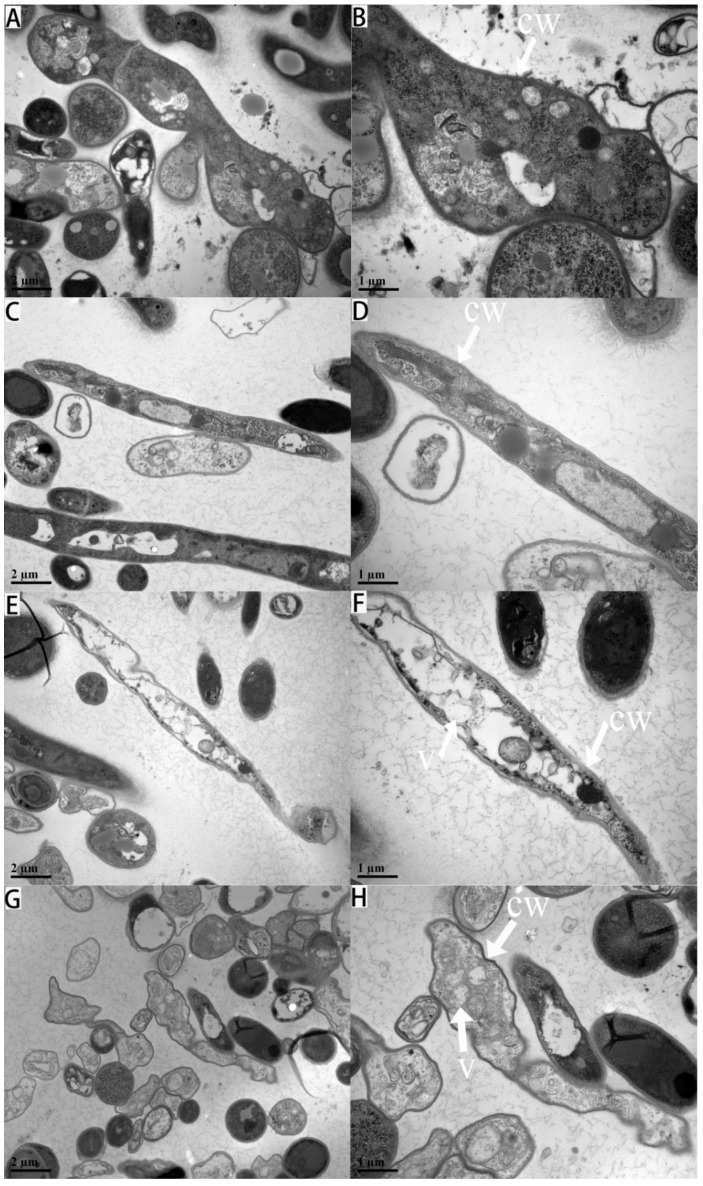
Ultrastructure of mycelial cells after 72 h treatment with different concentrations of citronellal: (**A**,**B**): CK; (**C**,**D**): 85 mg/L; (**E**,**F**): 134 mg/L; (**G**,**H**): 218 mg/L. The arrows in the figure indicate cell wall (cw) and vacuole (v).

**Figure 7 jof-08-01310-f007:**
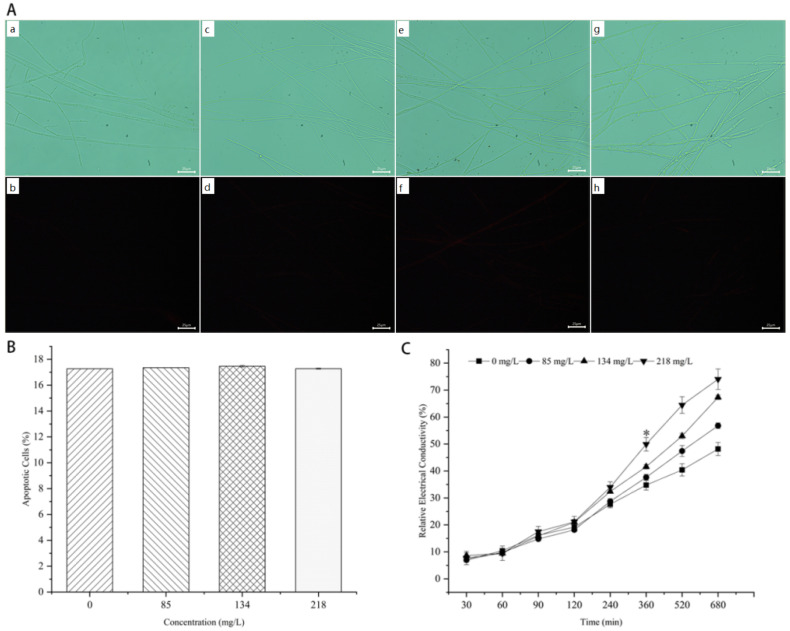
Effect of citronellal on the cell membrane of *M. oryzae*. (**A**,**B**): PI fluorescence and nonfluorescent effects of mycelia after treatment with citronellal for 72 h: (**a**,**b**): CK; (**c**,**d**): 85 mg/L; (**e**,**f**): 134 mg/L; (**g**,**h**): 218 mg/L. (**C**): Effect of citronellal on mycelial cell membrane permeability of *M. oryzae*.

**Figure 8 jof-08-01310-f008:**
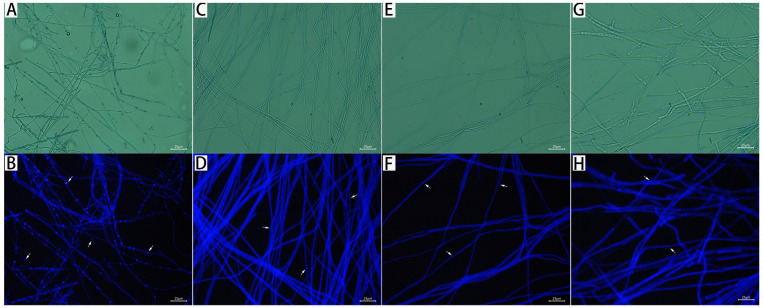
CFW fluorescence effect and nonfluorescent of mycelia after treatment with citronellal for 72 h. (**A**,**B**): CK; (**C**,**D**): 85 mg/L; (**E**,**F**): 134 mg/L; (**G**,**H**): 218 mg/L.

**Figure 9 jof-08-01310-f009:**
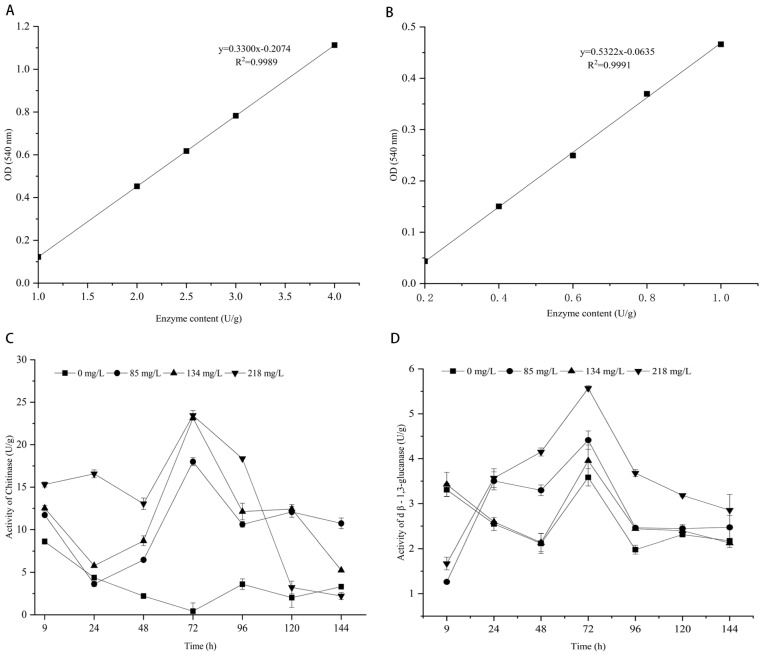
Enzyme activity of mycelium in *M. oryzae* treating with citronellal. Standard curve of chitinase (**A**) and β-1,3-glucanase (**B**); Change trend of chitinase (**C**) and β-1,3-glucanase (**D**) activity.

**Figure 10 jof-08-01310-f010:**
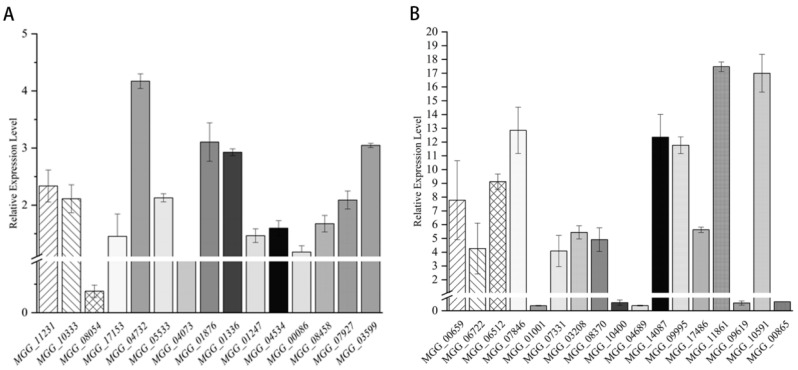
Relative expression level of gene of chitinase (**A**) and β-1,3-glucanase (**B**).

**Figure 11 jof-08-01310-f011:**
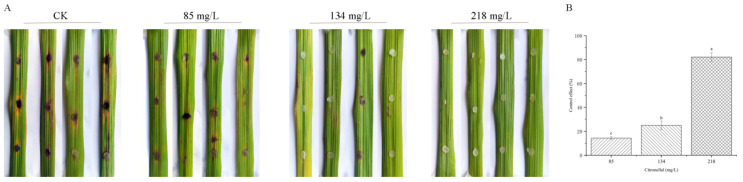
Indoor control effect of citronellal on rice blast. **(A)** Rice leaves inoculated with *M. oryzae*. **(B)** Control effect. Error bars indicate means ± SD (*n* = 3).

**Table 1 jof-08-01310-t001:** Primer design for RT-qPCR.

Gene	Primer Name	Forward Primer (5′–3′)	Reverse Primer (5′–3′)
actin	*MGG*_03982	CTGGCACCGTCGTCGATGAAG	AAGGTCCGCTCTCGTCGTACTC
42 kDa endochitinase	*MGG*_11231	GGCAACTCCAGCGAGATATTTCCC	GGTATTCCCAGTCAACGTCAATCCC
class III chitinase	*MGG*_10333	CCTCGCCTGTGCCTTCCATTTG	GTGACCAACAAAGAAGACGCCAAAC
chitinase 1	*MGG*_08054	CTCATCCCAACAAACTCG	GCGTCAAGCCACTCATAT
putative uncharacterized protein	*MGG*_17153	ACCCGCTGGATGGTGTACTACG	CGACGAGTTGAGGAACAAGTCTGAC
acidic mammalian chitinase	*MGG*_04732	AAGGGCAAGTACATCACCGACAAC	CATGGCATCGACGACAATGTTCTTG
chitinase 1	*MGG*_05533	CGTCGCCTATGTGACCAACTGG	GGCAAGAATATCCGCAAACGCATAC
class III chitinase	*MGG*_04073	CATCACGCTCAACGACCACCAC	CCCTCCCAGCATCCCGAGAATC
chitinase 3	*MGG*_01876	CGTCGCTCTCCTAGCAACAACAG	GAGTGCGTCGTCTGGAAGTAGATG
bacteriodes thetaiotaomicron symbiotic chitinase	*MGG*_01336	CGGAGGAAACGTGACCACAGAC	TTGACAGCCAGTGCCACAATGAG
chitinase 1	*MGG*_01247	AGAACCGTTACTGTCACCGTTGATG	GGACTTGTGCCAGAGGGTATGATG
chitinase	*MGG*_04534	CCACCTGTCACTACAAGAGCGAATG	AAGCCAAACTCTGAGCAGCACAC
endochitinase	*MGG*_00086	CACCCATCTGCTTTATGCGTTTGC	TCTGTGGGGTAGTGCTTGGAGAC
chitinase 1 precursor	*MGG*_08458	CCCTCACCTCCCAGAAACA	TTGAGTAGGCTGCGATGGATA
endochitinase 1	*MGG*_07927	GTCGTTTCCTTCGCCCTGATAGTC	GTGGAGAAGATTGTGGTGGAGTGTC
acidic endochitinase SE2	*MGG*_03599	TTAGGCTGCTGACGGCTAGTCC	GTCAAACCTCACCGCCCGAATC

**Table 2 jof-08-01310-t002:** Primer design for RT-qPCR.

Gene	Primer Name	Forward Primer (5′–3′)	Reverse Primer (5′–3′)
glucan 1,3-beta-glucosidase	*MGG*_00659	CGACAAACATCCCACCTCAGACTC	AGAAACCGCCAGACCCAGACTC
1,3-beta-glucanosyltransferase gel2	*MGG*_06722	ATTTCAAGAAGGTTCAGGGCGTCTC	CGGGCAGCGTAAAGTTGTTGTTG
glucan 1,3-beta-glucosidase 2	*MGG*_06512	CCAGGACCGTTACCGCAACATC	ACCATTCTTCCGCACCAAGTCATAC
endoglucanase family 5 glycoside hydrolase	*MGG*_07846	CACCACTTCTTCACAGAGGCAGAC	GCTTCTTGATGACGGACGGGTTG
endo-1,3(4)-beta-glucanase 1	*MGG*_01001	ACCGAGGACTTGCCATCAATATGC	GTGAGGATGCCGATGTTGGTGTC
1,3-beta-glucanosyltransferase gel1	*MGG*_07331	CCGCCAAGATTGAGGATGCTGAG	TTGTTGCTGCCCTGGTTGCTAG
glycolipid-anchored surface protein 5	*MGG*_03208	TGGATGGGATGGAACTGGGATGG	TAAAGTGCCCGTGACCTCTCCTG
1,3-beta-glucanosyltransferase gel3	*MGG*_08370	GGGTCGTTGGTTAGGTGTTGGATAC	CCGCACCACTCGTAGATGTTGTAG
GPI-anchored cell wall beta-1,3-endoglucanase EglC	*MGG*_10400	GCAATCCAGGGCTTCAACTACGG	TACAGGCGGGCAGAGTTCCATC
glucan 1,3-beta-glucosidase	*MGG*_04689	CATCACCAAGAACCTGTCCACCAAG	TGTTGCTACCACCAGTGCTGTTTC
beta 1,3 exoglucanase	*MGG*_14087	GCAAGGCAGACGACACAGTAGC	AGGAGTGGTGGTGGTGGAGTTG
glucan 1,3-beta-glucosidase	*MGG*_09995	TTACCTTGTCAGCGGCACGATTC	GATGACTCCCAGCCCAAGAAACG
glucan 1,3-beta-glucosidase	*MGG*_17486	ACTTCTTCCGCTCAACTCGCAAC	GGTGTTCTCCTCAGTACGCATCTTG
1,3-beta-glucanosyltransferase gel4	*MGG*_11861	CGCAGTACGACCAGCACATCAAG	AGCACCGCCAATACCGTCAATG
GPI-anchored cell wall beta-1,3-endoglucanase EglC	*MGG*_09619	AGGCGAGCGGTCAGGATAATGG	ACCCAGTCTCCGTGACCCAAAC
endo-beta-1,3-glucanase	*MGG*_10591	CACTACGACGACCGCAGTCAATAC	GTCCTCCTGTTGTTGGTGGTGATG
1,3-beta-glucan synthase component FKS1	*MGG*_00865	CGACGCCAACCAGGACAACTATC	GACGGCATTCTTGACACCAGGAG

## Data Availability

This study did not report any data.

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
