# Peer review of "Natural Product Citronellal can Significantly Disturb Chitin Synthesis and Cell Wall Integrity in Magnaporthe oryzae"

_jof, 2022, doi:10.3390/jof8121310_

Round 1
Reviewer 1 Report
Dear Colleagues.
I have some questions about your work:
First of all, I would like this article to be checked by a native English speaker. I am not one, but it seems to me that the authors use some words and verbal constructions incorrectly. The text is hard to understand in places.
The use of biologically active substances as fungicides is of great interest. Undoubtedly, this direction will reduce environmental damage. Similar compounds with antifungal activity have been isolated from some crops, in particular from garlic. The authors used citronal. Their choice is not very clear. Comments and suggestions to the authors in the attached file

Author Response
亲爱的审稿人
非常感谢您对我们稿件的关注和考虑, 请见附件!

Reviewer 2 Report
I have gone through the manuscript and has following suggestions
1. Method section need a through grammatical revision. It has given as protocol and not a what has done. Authors has also not made is clear that the method was followed with or without modifications.
2. SPSS software reference is missing
3. Instead of “referred to” in material section the authors can use “The method suggested by ….. was followed”
4. Line “Critical point drying, coating, and observation.” Is incomplete
5. Instead of trade name, author should name the active ingredient of “HEPENGBIO staining solution”
6. Most of the kits and regents are produced in China, and due to that the global audiecnce may face difficulty in accessing the kits protocols. So, authors should give the detail methodology
7. “inoculated M. oryzae cake” what it mean? No clear , please revise
8. “incidence of rice leaves was counted” What is counted?
9. “necrotic cells lose membrane integrity, while necrotic cells can enter cell and bind to DNA” need a revision for clarity
10. At mRNA level expression level indicates that citronellal might have affected the subsequent synthesis of β-1,3-glucan, but the enzyme assay couldn’t prove it. The author should discuss the reason.
Author Response
Dear Reviewer
Thanks very much for your attention and consideration to our manuscript. Please see the attachment!

Round 2
Reviewer 1 Report
Dear colleagues. There is still a question about the Figure 7, please check

Author Response
Dear Reviewr
Thank you very much for your attention and thinking of our manuscript! The attachment is my detailed explanation. Please see the attachment!
